# External validation of the priapism impact profile in a Jamaican cohort of patients with sickle cell disease

**Belinda F. Morrison**[1]*⊕, **Wendy Madden**[2]⊕, **Monika Asnani**[2]⊕, **Ayodeji Sotimehin**[3]⊕, **Uzoma Anele**[3]⊕, **Yuezhou Jing**[3]⊕, **Bruce J. Trock**[3]⊕, **Arthur L. Burnett**[3]⊕

**1** Department of Surgery, University of the West Indies, Mona, Kingston, Jamaica, **2** Caribbean Institute for Health Research- Sickle Cell Unit, University of the West Indies, Mona, Kingston, Jamaica, **3** The James Buchanan Brady Urological Institute of Department of Urology, The Johns Hopkins University School of Medicine, Baltimore, Maryland, United States of America

⊕ These authors contributed equally to this work.
* Belinda.morrison02@uwimona.edu.jm

**Data Availability Statement:** The data has been uploaded as a supplemental file.

**Funding:** The authors received no specific funding for this work.

## Abstract

### Background

Priapism impairs quality of life and has a predilection for males with sickle cell disease (SCD). The Priapism Impact Profile (PIP) is a novel 12-item instrument designed to measure general health-related impact of priapism. The aim of the study was to evaluate the validity and reliability of the PIP in a Jamaican cohort of SCD patients experiencing priapism.

### Methods

One hundred SCD patients with a history of priapism were recruited from a sickle cell clinic in Kingston, Jamaica and administered the PIP questionnaire. Patients rated each item of the PIP for clarity and importance. Statistical testing was employed to evaluate the psychometric performance of the PIP. Content validation was assessed based on patient descriptive rating of the items based on clarity, and importance and criterion-oriented validity were assessed by evaluating the PIP's ability to distinguish between patient subgroups. Test-retest repeatability was assessed in 20 of the 100 patients.

### Results

Patients were stratified into active (54) and remission (46) priapism groups based on their experience of priapism within the past year. Patients in the active priapism group were younger (p = 0.011), had a shorter duration of disease (p = 0.023), and had more frequent priapism episodes (p = 0.036) than the remission group. PIP questionnaire scores differed significantly with respect to priapism activity (p < 0.001) and prevalence of erectile dysfunction (p < 0.05) but not by priapism severity (p = 0.62). The PIP questionnaire had good content validity, with questions rated as having medium or high clarity and importance by an average of 82.8% and 69.2% of patients, respectively.

**Competing interests:** The authors have declared that no competing interests exist.

## Conclusion

The PIP questionnaire was successfully validated in a Jamaican cohort of SCD patients and adequately discriminated patients with active priapism from those in remission. The instrument may be utilized in routine clinical management of patients with SCD-associated priapism. Further clinical investigations are warranted in other populations.

## Introduction

Priapism, defined as a persistent penile erection continuing beyond or unrelated to sexual stimulation, is an uncommon disorder [1]. Whereas the incidence of priapism in the general population is quite low, the disease disproportionately affects the population of males with sickle cell disease (SCD) [2,3]. Ischemic priapism is predominantly seen in SCD and is typified by reduced or absent corporal flow. The cumulative incidence of priapism in SCD by age 40 years is as high as 60% [3]. The condition has a predilection for young males, with the peak age of onset in the late teens [4,5]. The probability of a male with SCD experiencing priapism by age 20 years is 89% [6]. Nationwide studies in the United States report that among males with priapism who present to the emergency department, those with SCD are more likely to be hospitalized and at a younger age than those with non-SCD related priapism [7–9]. The burden of priapism in SCD is even greater in Jamaica where 10% of the population carries the defective sickle hemoglobin gene [10].

Priapism is a potentially debilitating complication in SCD and is often associated with frequent episodes of bothersome painful nocturnal erections, frequently referred to as stuttering priapism [1]. The condition often leads to erectile dysfunction (ED) in as many as 47.5% of patients because of recurrent episodes in association with progressive corporal fibrosis [11–13]. Due to its repeated and prolonged episodes of high-pressure vascular distension of the corporal bodies, priapism may also result in conspicuous disfigurement of the penis called megalophallus. All these factors may lead to impairment in overall quality of life in these patients. The psychological impact of priapism in SCD includes despair, embarrassment and isolation [14]. Despite the obvious psychological effects of priapism in these patients, very little is presently done to assess and comprehensively monitor their health-related outcomes.

The priapism impact profile (PIP) questionnaire was pioneered as the first instrument to assess the general health-related impact of priapism [15]. It was originally designed and tested in 54 adult patients with a history of priapism (36 males with SCD and 18 males without SCD) attending the urology and hematology clinics at the Johns Hopkins Hospital between 2011 and 2014. The preliminary study performed well with respect to reliability and validity, distinguishing between patients with active and remitted disease, and those with severe erectile dysfunction. We wished to test the hypothesis that the PIP is a scientifically valid and clinically useful patient reported outcome instrument to ascertain the patient perspective of having priapism by performing an external validation study. We therefore proposed to externally validate the PIP questionnaire in a cohort of patients with SCD having both active and remitted priapism attending a clinic in Kingston, Jamaica.

## Materials and methods

### Questionnaire composition

The PIP questionnaire consists of 12 items in 3 domains: Quality of life (QoL), sexual function (SF) and physical wellness (PW). Each item has a 7-point scale ranging from 1 (absent) to 7

(very extreme) leading to a total possible score ranging from 12–84. Higher scores indicate greater severity within each domain. Each question was answered based on their experiences over the prior 2 week period to ensure good patient recall.

## Population description and enrollment

All patients were recruited from the Sickle Cell Unit, a large outpatient clinic serving only patients with sickle cell diseases and located at the University of the West Indies, Mona, Kingston, Jamaica, from May 1, 2015- June 30, 2017. The clinic is the only one of its kind in the English-speaking Caribbean. Patients were recruited and consented after presenting for routine health maintenance visits. Inclusion criteria were adult males (≥18 years) with self-reports of priapism and hemoglobin electrophoretic-proven diagnosis of SCD [16]. Patients were excluded if they were unable to comprehend and self-complete questionnaires or if they presented with acute illnesses at the time of clinic visits including priapism requiring urgent surgical or medical ablative therapies such as penile shunt surgery, penile prosthesis implantation, penile vascular surgery, or androgen deprivation.

## Study procedures

Patients self-completed questionnaires within 15 minutes in a private room in the clinic. The patients were next instructed to evaluate the instrument by descriptively rating each of the items for clarity (i.e. understanding) and importance as low, medium or high. All patients completed the 5-item International Index of Erectile Function (IIEF) questionnaire which objectively assessed erectile function [17]. The recruiter then collected additional demographic information and details on priapism history (i.e. age of onset, frequency and duration of priapism episodes, and prior treatments). One in 5 patients were randomly asked to return within 2–4 weeks to assess test-retest reliability of the instrument. The questionnaire was pre-tested on 30 eligible patients to ensure understanding and acceptability of the questions. Once it was deemed that the questionnaire was understandable and acceptable, it was administered to the remainder of the recruited patients.

## Outcome measurements

As in the original study, criterion-oriented validity was assessed by evaluating the PIP's ability to distinguish between patient subgroups based on priapism activity (≥1 year or < 1 year since last episode), priapism severity (duration of episodes), and ED prevalence. Active priapism was defined as having at least one priapism episode without spontaneous remission within the past year; spontaneous remission was defined as no active priapism episodes at least within the past year. The purpose of this distinction was to evaluate the performance of the instrument comparing findings when the disorder was active or in remission. Priapism of high severity was defined as priapism episodes lasting > 2 hours while low severity was defined as priapism episodes lasting ≤2 hours. ED was classified as present (mild to severe) and not present (none) based on the specified ranges of the IIEF questionnaire scores [17]. Content validation was assessed based on patient descriptive rating of the items using terms such as high, medium or low importance or high, medium or low clarity.

## Statistical analysis

The study sample comprises 100 patients, representing all eligible patients during the time frame of the study. Means ± standard deviations (SD) as well as medians with interquartile range (IQR) were calculated. Continuous data were compared using the Wilcoxon rank sum

nonparametric test. Categorical data were compared using the Chi Square or Fisher's Exact test as appropriate. Internal consistency coefficients for the total PIP questionnaire and the 3 subscales (domains) were generated using Cronbach's α. Cronbach's α was calculated with acceptable values ranging from 0.70 to 0.95. Test-retest analysis was performed using paired t-tests and McNemar Chi-square test. With 100 patients the study had 84% power to detect relatively small effect sizes (i.e. good precision) ≥ 0.6 [18]; power ≥80% to detect differences in proportions ≥27 percentage points [19]; and power = 86% to detect Chronbach's α>0.7 compared to a null hypothesis = 0.5 [20]. A p value of < 0.05 was considered statistically significant. Statistical analysis was performed using SAS v9.4 (SAS Institute, Cary, NC).

## Ethics statement

This study was conducted in accordance with the Declaration of Helsinki (as revised in 2013). This study was approved by the Ethics Committee, University of the West Indies, Mona, Kingston, Jamaica (IRB No. ECP 20, 16/17), and individual consent was obtained from each patient.

## Results

### Demographics and priapism characteristics

A total of 100 sickle cell disease patients with a history of priapism completed the questionnaire. Of these, 86.8% had Hemoglobin SS disease, 6.6% had Hemoglobin Sβ-thal disease, and 6.6% had Hemoglobin SC disease.

Patients were stratified into active (54) and remission (46) priapism groups based on whether they experienced a priapism episode within the past year. Patients in the active priapism group were younger (p = 0.011), had a shorter duration of disease (p = 0.023), and had more frequent priapism episodes (p = 0.036) than the remission group (Table 1). No

**Table 1. Demographic and clinical characteristics of active and remission priapism patients.**

| | Active Priapism (n = 54) | Remission Priapism (n = 46) | P-Value |
|---|---|---|---|
| Median Age, yrs (IQR) | 26.5 (22–33) | 32 (24–42) | 0.011 |
| Marital Status, n (%) | | | 0.192 |
| Married | 4 (10.3) | 8 (21.1) | |
| Unmarried | 35 (89.7) | 30 (78.9) | |
| Hemogloblin Status, n (%) | | | 0.259 |
| HbSS | 38 (90.5) | 28 (82.4) | |
| HbSβ | 3 (7.1) | 2 (5.9) | |
| HbSC | 1 (2.4) | 4 (11.8) | |
| Priapism History, yrs median (IQR) | | | |
| Age of Onset | 17 (14–22) | 20.0 (14.5–25) | 0.186 |
| Overall Duration | 7.5 (4–14) | 10.5 (6–22) | 0.023 |
| Episode Frequency, n (%) | | | 0.036 |
| Daily (1–7 episodes/wk) | 20 (43.5) | 8 (21.6) | |
| Monthly (<4 episodes per month) | 26 (56.5) | 29 (78.4) | |
| Episode Duration, n (%) | | | 0.867 |
| ≤2 hrs ("very minor") | 30 (58.8) | 21 (55.3) | |
| 2–5 hrs ("minor") | 19 (37.3) | 16 (42.1) | |
| >5 hrs ("major") | 2 (3.9) | 1 (2.6) | |
| Erectile Dysfunction, n (%) | 19 (38.8) | 22 (48.9) | 0.323 |

**Table 2. A Priapism Impact Profile (PIP) questionnaire criterion-oriented validity of domains based on comparison of patient subgroups.**

| Patient Subgroups | Priapism Activity, median (IQR) | | | Priapism Severity, median (IQR) | | | Erectile Dysfunction, median (IQR) | | |
|---|---|---|---|---|---|---|---|---|---|
| | Active (n = 54) | Remission (n = 46) | P value | High (n = 3) | Low (n = 86) | P value | Present (n = 41) | Absent (n = 59) | P value |
| Total PIP Score | 37 (25–48) | 21 (16–27) | 0.0003 | 30 (19–48) | 27 (18–42) | 0.623 | 36 (19–50) | 23 (18–33) | 0.026 |
| Quality of Life | 13 (9–18) | 7 (6–10) | 0.0001 | 10 (7–19) | 10 (6–16) | 0.589 | 13 (8–19) | 8 (6–11) | 0.004 |
| Sexual Function | 10 (6–15) | 5 (5–10) | 0.005 | 9 (5–16) | 7 (5–13) | 0.220 | 11 (5–18) | 6 (5–10_ | 0.008 |
| Physical Wellness | 12 (7–15) | 7 (3–10.5) | 0.001 | 10 (6–15) | 9 (6–14) | 0.556 | 10.5 (5–15) | 9 (5.5–12) | 0.147 |

significant differences were seen between the two groups for age of onset of priapism, priapism severity or ED rates.

## Questionnaire outcomes

Patients uniformly completed the questionnaire within 15 minutes with no difficulties. For criterion-oriented validity, we demonstrated the ability of both the total and subscale dimensions of this instrument to distinguish between subgroups, finding uniformly higher scores for active priapism and existent ED relative to their converse conditions, but no statistically significant difference for priapism severity. The active priapism group had a significantly higher total PIP score, and higher QoL, SF and PW scores than that of the remission group (all $p \leq 0.005$) (Table 2).

Considering each of the individual PIP questions separately, the instrument was able to distinguish between subgroups in the entire sample with most questions exhibiting significantly higher scores for active priapism and presence of erectile dysfunction but not priapism severity (Table 3).

See S1 File for the original **Priapism Impact Profile Questionnaire**

The PIP questionnaire had fairly good content validity, with patients assigning medium or high "clarity" or "importance" by an average of 82.8% and 69.2%, respectively, over all 12 questions (Table 4). The PIP questionnaire had excellent reliability. The item-total Cronbach's α reliability coefficient for the total PIP score was 0.926. Individual PIP domains had α coefficients of 0.8577 for QoL, 0.925 for SF, and 0.778 for PW (Table 5).

Twenty patients completed the questionnaires a second time at an interval up to 1 year after initial completion. We compared retest vs. initial responses with respect to the individual 12 questions (S1 Table in S1 File); the Total PIP score and QOL, SF, and PW domains (S2

**Table 3. PIP questionnaire criterion-oriented validity of questions 1–12 based on comparison of patient subgroups.**

| Items | Q1 | Q2 | Q3 | Q4 | Q5 | Q6 | Q7 | Q8 | Q9 | Q10 | Q11 | Q12 |
|---|---|---|---|---|---|---|---|---|---|---|---|---|
| Priapism Activity, median (IQR) | | | | | | | | | | | | |
| Active | 4 (2–6) | 4 (2–5) | 2 (1–3) | 3 (1–5) | 2 (1–4) | 2 (1–3) | 2 (1–4) | 1 (1–2) | 1 (1–3) | 4 (2–6) | 5 (3–6) | 2 (1–4) |
| Remission | 4 (2–5) | 1 (1–3) | 1 (1–2) | 1 (1–2) | 1 (1–2) | 1 (1–2) | 1 (1–2) | 1 (1–2) | 1 (1–3) | 2 (1–4) | 3 (1–6) | 1 (1–3) |
| p value | 0.172 | <0.0001 | 0.005 | 0.0005 | 0.010 | 0.057 | 0.003 | 0.578 | 0.441 | 0.002 | 0.002 | 0.157 |
| Priapism Severity, median (IQR) | | | | | | | | | | | | |
| High | 4 (2–6) | 3 (1–6) | 2 (1–3) | 2 (1–5) | 2 (1–4) | 2 (1–3) | 2 (1–3) | 1 (1–2) | 1 (1–3) | 3 (1–5) | 6 (2–6) | 1 (1–4) |
| Low | 4 (2–6) | 2 (1–5) | 1 (1–3) | 2 (1–3) | 1 (1–3) | 1 (1–2) | 1 (1–3) | 1 (1–1) | 1 (1–3) | 3 (1–6) | 4 (3–6) | 1 (1–3) |
| p value | 0.908 | 0.510 | 0.488 | 0.307 | 0.642 | 0.064 | 0.306 | 0.113 | 0.646 | 0.409 | 0.165 | 0.438 |
| Erectile Dysfunction, median (IQR) | | | | | | | | | | | | |
| Present | 5 (3–6) | 4 (2–6) | 2 (1–4) | 3 (1–5) | 2 (1–4) | 2 (1–4) | 1 (1–2) | 1 (1–2) | 2 (1–4) | 3 (1–6) | 6 (2–7) | 2 (1–4) |
| Absent | 3 (2–4) | 2 (1–4) | 1 (1–2) | 2 (1–3) | 1 (1–2) | 1 (1–2) | 1 (1–2) | 1 (1–1) | 1 (1–2) | 3 (1–4) | 4 (2–6) | 1 (1–3) |
| p value | 0.007 | 0.008 | 0.0002 | 0.039 | 0.044 | 0.047 | 0.125 | 0.001 | 0.0004 | 0.295 | 0.167 | 0.131 |

**Table 4. Content validity utilizing patient evaluation of Item importance and clarity.**

| | Number Rating Item as Medium or High/Number of Respondents to the Item | | | | | | | | | | | |
| | Q1 | Q2 | Q3 | Q4 | Q5 | Q6 | Q7 | Q8 | Q9 | Q10 | Q11 | Q12 |
|---|---|---|---|---|---|---|---|---|---|---|---|---|
| Importance (%) | 89/99 (89.9) | 75/96 (78.1) | 61/97 (62.9) | 68/97 (70.1) | 64/97 (66.0) | 58/96 (60.4) | 62/96 (64.6) | 59/96 (61.5) | 61/97 (62.9) | 70/95 (73.7) | 81/97 (83.5) | 51/91 (56.0) |
| Clarity (%) | 90/95 (94.7) | 83/94 (88.3) | 76/97 (78.4) | 77/95 (81.1) | 78/93 (83.9) | 77/93 (82.8) | 58/91 (63.7) | 71/93 (76.3) | 82/93 (88.2) | 81/91 (89.0) | 86/93 (92.5) | 64/ 87 (73.6) |

Table in S1 File); and the proportion who assigned medium-to-high importance or medium-to-high clarity to each of questions 1–12 (S3 and S4 Tables in S1 File, respectively). No comparisons were statistically significant with the exception of the question 3 importance, where medium-to-high importance was assigned by 90% of the 20 patients in their initial responses vs. 65% of retest responses (p = 0.05). Over all 12 questions, the average percentage of medium-to-high importance was 79.7% in initial response, and 77.5% in retest response (S5 Table in S1 File), and medium-to-high clarity was 89.9% in initial response and 92.8% in retest response (S6 Table in S1 File).

## Discussion

In this external validation study, the PIP, a novel instrument developed to assess the general health-related impact of priapism in males, proved to be excellent with respect to validity and reliability among this Jamaican population with SCD. The generation of such a patient reported outcome tool is a milestone in the management of SCD males with priapism considering the burden of the disease in these males and the dearth of assessments of the quality of life impact associated with priapism.

With respect to criterion-oriented validity, the instrument was able to discriminate between patients with active and remitted disease, with the former having predictably higher scores. This was consistent with findings from the preliminary study [15]. Median total PIP scores in our study for active and remitted disease were 37 and 21, respectively compared to 45.5 and 21.5 in the preliminary study. The instrument was also able to significantly discriminate between patient groups based on erectile function. Patients with more severe ED had higher scores and greater bother, which was consistent with the preliminary study [15]. Most patients in the study had priapism episodes of low severity (≤2 hours duration), however, there was no difference in priapism severity scores between patients with active or remitted disease. Whereas in the preliminary study, the PIP was able to distinguish between patient groups based on disease severity score, we were not able to demonstrate this in our study. It is logical to believe that priapism episodes of greater than 2-hour duration would cause great distress

**Table 5. Cronbach's alpha reliability coefficients for PIP score, QoL, SF, and PW to demonstrate good item-domain/total instrument interrelatedness.**

| | Cronbach's Alpha coefficients |
|---|---|
| Quality of Life | 0.857 0.790–0.881 (removing Q4 causes largest decrease) |
| Sexual Function | 0.925 0.897–0.927 (removing Q5 causes largest decrease) |
| Physical Wellness | 0.778 0.609–0.761 (removing Q11 causes largest decrease) |
| Total PIP | 0.926 0.915–0.929 (removing Q3 causes largest decrease) |

due to prolonged episodes of pain. We suggest that patients in our study may have developed coping strategies which allowed them to have adjusted to the chronic pain leading to less distress and bother and lower PIP scores. Studies on patients with SCD with chronic pain have demonstrated that coping strategies were predictors of adjustment and influenced psychological distress [21]. Our study did not evaluate coping strategies in enrolled patients.

The content validity demonstrated in the original study was validated in our population of patients with SCD. All questions in our study were rated as having medium or high clarity and importance by 82.8% and 69.2% of patients, respectively. This compares favorably to 93% and 78% of patients, respectively, in the preliminary study [15]. Several questions, inclusive of questions 3, 6 and 12 were rated as having lower importance than in the original study. The differences in these responses could be attributed to cultural differences and communication norms between the study populations.

The instrument's internal consistency (Cronbach's α), which is a reliability measure of item intercorrelation, was 0.926 in our study. This was similar to the findings in the preliminary study where the Cronbach's α was 0.9 [15]. We defined an acceptable Cronbach's α as 0.7 to 0.95 and both studies have acceptable reliability measures. The test-retest reliability was excellent in our population. Unfortunately, some of the patients returned with delays of up to 1 year for retesting due to socio-economic factors.

Instruments have been used in the SCD population to assess quality of life in general [22–27]. The WHOQOL-Bref and the Short form-36 (SF-36) questionnaires which have applicability in diverse clinical settings are instruments that have been used to assess quality of life in SCD [22,27]. Other instruments such as the Adult Sickle Cell Quality of Life Measurement Information System (ASCQ-Me) and the Sickle Cell Impact Measurement Scale (SIMS) have also been used. Very few of the instruments used to assess quality of life in SCD are validated in this population [26]. The PIP is the only instrument that has been designed to primarily assess quality of life in priapism only. The PIP therefore serves as a new quality of life tool in the management of patients with priapism in SCD.

The PIP will be very useful in the clinical setting in any patient with recurrent priapism. It should ideally be used as a routine screen for all patients and those with high scores should be referred for psychological assessment and treatment. The tool will also be useful during treatment to assess the impact of treatment on quality of life, sexual function and physical wellness by observing changes in scores with time. Additionally, the PIP will give more insight in the natural history of priapism in sickle cell disease by observing changes in scores over time and during various active phases of the disease.

Several strengths of this study are noteworthy. The study comprises a rather large sample size of patients with SCD-associated priapism, which is uncommon in many literature reports for this disease state. In addition, given the strict inclusion criteria for our study population, the findings are informative for SCD- associated priapism differentiated from other causes of priapism. In addition, the external validation of the instrument in a culturally distinct population, differing from the original study population, is important. This validation suggests the comprehensiveness of this instrument and its potential universality in further use.

Several limitations require mentioning. The study represented a single-institution analysis and therefore results may not be generalizable to all patients with SCD. Due to the cross-sectional design of the study, the instrument was limited in assessing changes in QOL with time. Recall bias could be an important bias inherent in this type of study, considering that the data relied on patient recall. In addition, patients with acute complications of SCD were excluded from the study, hence the generalizability of our results to patients with acute illnesses is uncertain. Data on the presence of complications of SCD were not collected, hence the presence of these may have been confounders in the evaluation of QOL. In addition, the questionnaire was

restricted to adult males so we cannot comment on its performance in non-adults with priapism. We validated the PIP in a Jamaican population including men only with SCD- associated priapism, so validation is needed in a non-SCD population. We believe that a process of adaptation of the questionnaire in the future is prudent, considering the cultural differences between the two study populations. This process of adaptation will bear in mind culture discourse norms in Jamaica and may change the perception of the importance of some of the questions as well as the severity of priapism.

## Conclusion

In conclusion, our study presents the first external validation study of the novel PIP questionnaire using a population of men with SCD in Jamaica. The instrument demonstrated sound psychometric performance, suggesting that it offers practical utility in the management of priapism. Specifically, it may be used by the clinical practitioner for assessing the baseline health-related impact of priapism in the afflicted individual. It is conjectured that the instrument will prove useful in assessing the effectiveness of existing and novel therapies in the management of priapism. Ongoing investigation including clinical trials is warranted to determine its use in monitoring outcomes in priapism associated with SCD and possibly other etiologies of priapism.

## Supporting information

**S1 File. Priapism Impact Profile Questionnaire.**
(DOCX)

**S2 File.**
(XLSX)

## Author Contributions

**Conceptualization:** Belinda F. Morrison, Monika Asnani, Arthur L. Burnett.

**Data curation:** Belinda F. Morrison, Ayodeji Sotimehin, Uzoma Anele, Yuezhou Jing.

**Formal analysis:** Belinda F. Morrison, Yuezhou Jing, Bruce J. Trock, Arthur L. Burnett.

**Investigation:** Belinda F. Morrison.

**Methodology:** Belinda F. Morrison, Bruce J. Trock, Arthur L. Burnett.

**Project administration:** Wendy Madden, Arthur L. Burnett.

**Resources:** Belinda F. Morrison, Monika Asnani, Ayodeji Sotimehin, Uzoma Anele.

**Software:** Ayodeji Sotimehin, Uzoma Anele, Yuezhou Jing, Bruce J. Trock.

**Supervision:** Belinda F. Morrison, Wendy Madden, Monika Asnani, Arthur L. Burnett.

**Validation:** Belinda F. Morrison, Yuezhou Jing, Bruce J. Trock.

**Visualization:** Belinda F. Morrison, Arthur L. Burnett.

**Writing – original draft:** Belinda F. Morrison, Arthur L. Burnett.

**Writing – review & editing:** Belinda F. Morrison, Wendy Madden, Monika Asnani, Ayodeji Sotimehin, Uzoma Anele, Yuezhou Jing, Bruce J. Trock, Arthur L. Burnett.

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
