## [Decision Letter · Decision Letter 0]

4 Aug 2021

PONE-D-21-19408

External validation of the priapism impact profile in a Jamaican cohort of patients with sickle cell disease

PLOS ONE

Dear Dr. Morrison,

Thank you for submitting your manuscript to PLOS ONE. After careful consideration, we feel that it has merit but does not fully meet PLOS ONE’s publication criteria as it currently stands. Therefore, we invite you to submit a revised version of the manuscript that addresses the points raised during the review process.

We look forward to receiving your revised manuscript.

Kind regards,

Ambroise Wonkam

Academic Editor

PLOS ONE

2. Please note that in order to use the direct billing option the corresponding author must be affiliated with the chosen institute. Please either amend your manuscript to change the affiliation or corresponding author, or email us at plosone@plos.org with a request to remove this option.

Reviewers' comments:

Reviewer's Responses to Questions

**Comments to the Author**

1. Is the manuscript technically sound, and do the data support the conclusions?

Reviewer #1: Yes

Reviewer #2: Yes

Reviewer #3: Yes

2. Has the statistical analysis been performed appropriately and rigorously? 

Reviewer #1: Yes

Reviewer #2: Yes

Reviewer #3: Yes

3. Have the authors made all data underlying the findings in their manuscript fully available?

Reviewer #1: No

Reviewer #2: Yes

Reviewer #3: Yes

4. Is the manuscript presented in an intelligible fashion and written in standard English?

Reviewer #1: Yes

Reviewer #2: Yes

Reviewer #3: Yes

5. Review Comments to the Author

Reviewer #1: The study team is to be commended on conducting an important study to further validate an instrument to measure priapism. The approach and analysis to analyze the psychometric properties was excellent. I have a few suggestions to present the work more clearly.

Table 2 – flip the columns and rows

Table 3 – not helpful, can you publish the actual PIP questions? If yes, then I would consider as a separate table. As is this table is confusing and would need a legend. I see it is included as supplemental; I would include in the main manuscript.

Table 4 – also not helpful and not really clearly presented. It looks like there were several items that scored low in level of importance. Was there discussion about eliminating these items? This should be discussed in the discussion section.

Table 5 – can you include a more typical presentation of the factor analysis results? Internal consistency, were there specific items that look like they could have been eliminated? Since an item analysis wasn’t presented I am not sure.

Could the team sit down together and discuss how to revise the tables to maximize their presentation of the results?

Great results.

Discussion:

P1 – “Landmark”, can you tone this down a little bit?

Overall the discussion is acceptable, not exciting, but acceptable. Can you include a little more on how this scale could be useful in a clinical setting? Would this become a routine screen? What would be the result of the screen? Referral to who? How would this help patients? I know it is an important instrument and would be beneficial to healthcare providers, but perhaps a paragraph on incorporating into the clinic setting and how it would help improve male’s lives would be beneficial to the reader.

Reviewer #2: In their original article entitled: “External validation of the priapism impact profile in a Jamaican cohort of patients with sickle cell disease”, Morrison et al. evaluated the validity and reliability of the priapism impact profile (PIP) in assessing the general health-related impact of priapism in a cohort of patients with sickle cell disease (SCD). Priapism is a frequent complications of SCD, and is known to alter the quality of life and can lead to erectile function. Assessing the impact of priapism in the course of SCD patients is thus of high importance. The PIP was originally designed and tested only once at the Johns Hopkin hospital in the United States. An external validation of this tool and preferably in the different setting was much needed.

To assess the validity of the PIP, Morrison et al. administered the questionnaire to 100 SCD patients recruited from a sickle cell clinic in Kingston, Jamaica. Patients were divided into active and remission priapism groups. The PIP tool was able to distinguish between subgroups with respect to active priapism, presence of erectile dysfunction, but not for priapism severity. Also, the PIP questions were rated as having medium or high clarity and importance by an average of 82.8% and 69.2% of patients, respectively. Based on these results, the authors advocated for the possible use of the PIP for the clinical management of Priapism and for the assessment of priapism therapies. Morrison et al. used a comprehensive and reproducible method to collect and analyze data, their results are well organized and presented, and the conclusions they made derived directly from their findings. I thus recommend the article by Morrison et al. for publication in PLOS ONE, subject to addressing the following minor comments:

Comment 1: The aim for this study was to assess the validity and the reliability of the PIP tool. The authors emphasized more on the validity of the tool, but did not clearly stated whether the tool was reliable or not. Please the reliability of the tool should be clearly mentioned in the results section.

Comment 2: Also, the reliability score (Cronbach’s α) was higher in the original/preliminary study (0.9) as compared to that of the present study (0.78). The authors explained that this difference may possibly be due to some factors such as degree of religiousness, life setting (urban vs rural), and level of noise in the interview area. This also give credit to the effect of the difference with respect to study settings on the reliability of the PIP. The authors should discuss a possibly amendment/adaptation that can be done to the PIP tool for it to suit the Jamaican context.

Comment 3: The PIP tool was able to distinguish difference with respect to priapism severity in the preliminary study performed at the Johns Hopkin Hospital, while it was not able to do so in the present study. In addition to the coping mechanisms as mentioned by the authors, this difference of result may also be due to the cultural difference between the two study settings. Therefore, as mentioned in my previous comment, the authors should discuss a possible adaptation of the PIP instrument to their actual study setting.

Comment 4: P9/L195-196: The active priapism group had significantly higher quality of life score as compare to the remission group. Does this finding means that according to the PIP tool patients with active priapism have better quality of life as compared to patients in remission? This finding should be clarified and discussed by the authors as well.

Comment 5: the authors found that 69.2% of patients rated questions from the PIP as having medium to high importance. This means that nearly 30% of the participants think that some questions are not important. The authors should discuss how they think the questions that were rated as having a lower importance (Q3, Q6, Q12) can be edited/improved to suit the Jamaican context.

Reviewer #3: The sickle cell population has been misunderstood over 100 years in America despite its many potentially life threatening complications as well as morbidities that negatively affects quality of life. The authors have taken time to develop a much needed disease specific quality of life instrument to measure the effects of one unique but frequently encountered complication of priapism in this population. The PIP is a brief instrument and it measures quality of life , sexual function and physical Wellness they have defined and developed within the PIP instrument. The study measures its reliability responsiveness and to detect important changes within subgroups. The original PIP was developed without shortcuts using the standard 5 step QOL instrument development in 2011 tested throughout 2014 which showed that it was reliable and valid then. The authors took the time to do a an externally validation in 100 in collaboration with a large sickle cell center in Jamaica. A PIP questionnaire this cohort of adult patients with sickle cell disease having both active and remitted remittent priapism in Kingston Jamaica. Although the number of subjects evaluated are small compared to other conditions such as diabetes, cancer, this is a catastrophic complication of adults and males with sickle cell disease and very important to establish QOL instruments that can be utilized to assess with drug study development , routine care, and with third payer parties. resulted in 100 patients to successfully test PIP validity.

The authors conducted the research with purpose and it is very important that we document quality of life issues among individuals with sickle cell disease.

In the discussion section the authors appropriately pointed out the advantages and limitations of the study in that it is a single institution analysis the cronbach alpha was on the low end and that they did not evaluate various treatments and acute complications which could have included blood therapy which may have even modified their results. However the patients gave their input and It was important and lined up with how they felt about the appropriateness of the instrument questions . Also mentioned that the study was limited only to the adult male population but they noted this occurs also in younger pediatric and adolescent sickle cell populations. This disease specific instrument targeting a very narrow specific complication in sickle cell patients is much needed .There are new pipeline therapies for which some have already been FDA approved. As such, this type of instrument is much needed.

Minor suggestions:

1. Table 3. Label under the Items column needs to be consistent for the reader by subgroup.

Active vs.Remisssion under the severity median column and under erectile dysfunction. Or I maybe reading it wrong. Please clarify.

6. PLOS authors have the option to publish the peer review history of their article (what does this mean?). If published, this will include your full peer review and any attached files.

Reviewer #1: No

Reviewer #2: No

Reviewer #3: No

---

## [Author Response · Author response to Decision Letter 0]

27 Sep 2021

September 22, 2021

Dr. Ambrose Wonkam

Academic Editor

PLOS One

Dear Dr. Wonkam,

RE: PONE-D-21-19408 External validation of the priapism impact profile in a Jamaican cohort of patients with sickle cell disease

Thank you for reviewing the submitted manuscript and considering our work for publication. We note the comments of the reviewers and seek to address each individually in the document herein. Please see our responses to each comment below.

Reviewer 1

Reviewer #1: The study team is to be commended on conducting an important study to further validate an instrument to measure priapism. The approach and analysis to analyze the psychometric properties was excellent. I have a few suggestions to present the work more clearly.

1. Table 2 – flip the columns and rows

Response

Thank you for your comments. The table was amended (See table 2 in results).

2. Table 3 – not helpful, can you publish the actual PIP questions? If yes, then I would consider as a separate table. As is this table is confusing and would need a legend. I see it is included as supplemental; I would include in the main manuscript.

Response

The PIP questionnaire was submitted with the original documents but will be uploaded again with the resubmission. A footnote will be included below the table directing readers to the S1 File with the PIP questionnaire. The table has also been reformatted to be reader-friendly.

3. Table 4– also not helpful and not really clearly presented. It looks like there were several items that scored low in level of importance. Was there discussion about eliminating these items? This should be discussed in the discussion section.

Response

 Table 4 has been reformatted so that it is reader-friendly. The authors do not believe that the items that scored lower should be eliminated. We instead believe it is prudent that a process of adaptation of the questionnaire may be carried out considering cultural norms in Jamaica. This may allow the questionnaire to be more understandable for patients. (Please see lines Paragraph 1, page 20 of the discussion)

4. Table 5 – can you include a more typical presentation of the factor analysis results? Internal consistency, were there specific items that look like they could have been eliminated? Since an item analysis wasn’t presented I am not sure.

Response

 Based on the Reviewer’s comment Table 5 has been revised to show the range for the change in Cronbach’s alpha associated with deleting each of the component questions one at a time as well as the overall Cronbach’s alpha for each domain. None of the items when deleted gave a large enough change to Cronbach’s alpha to justify eliminating it. 

5. Could the team sit down together and discuss how to revise the tables to maximize their presentation of the results?

Response

We thank the reviewers for these comments and have made changes to the tables for ease of reading. Table 5 had corrections made to the content. We apologize as the results erroneously presented earlier represented analysis of early incomplete data. We herein report the correct analyses with the finalized data for Table 5.

6. Great results.

Discussion:

P1 – “Landmark”, can you tone this down a little bit?

Response

We have replaced the term “landmark” with “milestone.” The authors believe that the novelty of the instrument, being the first of its kind and its potential use a tool in management of priapism warrants its description as a milestone achievement. (Please see paragraph 1, page 17)

7. Overall the discussion is acceptable, not exciting, but acceptable. Can you include a little more on how this scale could be useful in a clinical setting? Would this become a routine screen? What would be the result of the screen? Referral to who? How would this help patients? I know it is an important instrument and would be beneficial to healthcare providers, but perhaps a paragraph on incorporating into the clinic setting and how it would help improve male’s lives would be beneficial to the reader.

Response

 The PIP will be very useful in the clinical setting in any patient with recurrent priapism. It should ideally be used as a routine screen for all patients and those with high scores should be referred for psychological assessment and treatment. The tool will also be useful during treatment to assess the impact of treatment on quality of life, sexual function and physical wellness by observing changes in scores with time. Additionally, the PIP will give more insight in the natural history of priapism in sickle cell disease by observing changes in scores over time and during various active phases of the disease. (Please see paragraph 2, page 19) 

Reviewer 2

In their original article entitled: “External validation of the priapism impact profile in a Jamaican cohort of patients with sickle cell disease”, Morrison et al. evaluated the validity and reliability of the priapism impact profile (PIP) in assessing the general health-related impact of priapism in a cohort of patients with sickle cell disease (SCD). Priapism is a frequent complication of SCD, and is known to alter the quality of life and can lead to erectile dysfunction. Assessing the impact of priapism in the course of SCD patients is thus of high importance. The PIP was originally designed and tested only once at the Johns Hopkin hospital in the United States. An external validation of this tool and preferably in the different setting was much needed.

To assess the validity of the PIP, Morrison et al. administered the questionnaire to 100 SCD patients recruited from a sickle cell clinic in Kingston, Jamaica. Patients were divided into active and remission priapism groups. The PIP tool was able to distinguish between subgroups with respect to active priapism, presence of erectile dysfunction, but not for priapism severity. Also, the PIP questions were rated as having medium or high clarity and importance by an average of 82.8% and 69.2% of patients, respectively. Based on these results, the authors advocated for the possible use of the PIP for the clinical management of Priapism and for the assessment of priapism therapies. Morrison et al. used a comprehensive and reproducible method to collect and analyze data, their results are well organized and presented, and the conclusions they made derived directly from their findings. I thus recommend the article by Morrison et al. for publication in PLOS ONE, subject to addressing the following minor comments:

Comment 1: The aim for this study was to assess the validity and the reliability of the PIP tool. The authors emphasized more on the validity of the tool, but did not clearly stated whether the tool was reliable or not. Please the reliability of the tool should be clearly mentioned in the results section.

Response

The reliability measure (Cronbach’s alpha) is described in reference to Table 5 – please see lines 195-196. An addition was made to the results section stating that the PIP questionnaire had excellent reliability.

Comment 2: Also, the reliability score (Cronbach’s α) was higher in the original/preliminary study (0.9) as compared to that of the present study (0.78). The authors explained that this difference may possibly be due to some factors such as degree of religiousness, life setting (urban vs rural), and level of noise in the interview area. This also give credit to the effect of the difference with respect to study settings on the reliability of the PIP. The authors should discuss a possibly amendment/adaptation that can be done to the PIP tool for it to suit the Jamaican context.

Response

The authors again apologize for the incorrect Cronbach’s alpha score that was reported in the initial submission of this manuscript (please see response #5 to Reviewer 1). This result was based on analysis of older incomplete data. The current value is 0.926 which is acceptable and very consistent with the prior study.

Comment 3: The PIP tool was able to distinguish difference with respect to priapism severity in the preliminary study performed at the Johns Hopkin Hospital, while it was not able to do so in the present study. In addition to the coping mechanisms as mentioned by the authors, this difference of result may also be due to the cultural difference between the two study settings. Therefore, as mentioned in my previous comment, the authors should discuss a possible adaptation of the PIP instrument to their actual study setting.

Response

The authors agree that cultural differences between the study populations could account for the differences in the perception of severity of priapism between the two groups. We agree that a process of adaptation of the questionnaire should be considered for the future; basing the adaptation on cultural discourse norms. (Please see paragraph 1, page 20 of the discussion)

Comment 4: P9/L195-196: The active priapism group had significantly higher quality of life score as compare to the remission group. Does this finding mean that according to the PIP tool patients with active priapism have better quality of life as compared to patients in remission? This finding should be clarified and discussed by the authors as well.

Response

The higher quality of life score in the active priapism cases translated to poorer quality of life and greater distress in these patients. Please see the explanation in the methods, lines 91-93- “Higher scores indicate greater severity within each domain.” 

Comment 5: the authors found that 69.2% of patients rated questions from the PIP as having medium to high importance. This means that nearly 30% of the participants think that some questions are not important. The authors should discuss how they think the questions that were rated as having a lower importance (Q3, Q6, Q12) can be edited/improved to suit the Jamaican context.

Response

The authors agree that cultural differences between the study populations could account for the differences in the perception of severity of priapism between the two groups. We agree that a process of adaptation of the questionnaire should be considered for the future; basing the adaptation on cultural discourse norms. (Please see paragraph 1, page 20 of the discussion)

Reviewer 3

The sickle cell population has been misunderstood over 100 years in America despite its many potentially life-threatening complications as well as morbidities that negatively affects quality of life. The authors have taken time to develop a much-needed disease specific quality of life instrument to measure the effects of one unique but frequently encountered complication of priapism in this population. The PIP is a brief instrument and it measures quality of life, sexual function and physical Wellness they have defined and developed within the PIP instrument. The study measures its reliability responsiveness and to detect important changes within subgroups. The original PIP was developed without shortcuts using the standard 5 step QOL instrument development in 2011 tested throughout 2014 which showed that it was reliable and valid then. The authors took the time to do a an externally validation in 100 in collaboration with a large sickle cell center in Jamaica. A PIP questionnaire this cohort of adult patients with sickle cell disease having both active and remitted remittent priapism in Kingston Jamaica. Although the number of subjects evaluated are small compared to other conditions such as diabetes, cancer, this is a catastrophic complication of adults and males with sickle cell disease and very important to establish QOL instruments that can be utilized to assess with drug study development, routine care, and with third payer parties resulted in 100 patients to successfully test PIP validity.

The authors conducted the research with purpose and it is very important that we document quality of life issues among individuals with sickle cell disease.

In the discussion section the authors appropriately pointed out the advantages and limitations of the study in that it is a single institution analysis the cronbach alpha was on the low end and that they did not evaluate various treatments and acute complications which could have included blood therapy which may have even modified their results. However the patients gave their input and It was important and lined up with how they felt about the appropriateness of the instrument questions. Also mentioned that the study was limited only to the adult male population but they noted this occurs also in younger pediatric and adolescent sickle cell populations. This disease specific instrument targeting a very narrow specific complication in sickle cell patients is much needed. There are new pipeline therapies for which some have already been FDA approved. As such, this type of instrument is much needed.

Minor suggestions:

Comment 1. Table 3. Label under the Items column needs to be consistent for the reader by subgroup.

Active vs. Remisssion under the severity median column and under erectile dysfunction. Or I maybe reading it wrong. Please clarify.

Response

Table 3 presents the criterion-oriented validity of questions 1-12 based on subgroups of priapism activity (active and remission), priapism severity (high and low) and erectile dysfunction (absent and present).

---

## [Editor Report · Decision Letter 1]

30 Sep 2021

External validation of the priapism impact profile in a Jamaican cohort of patients with sickle cell disease

PONE-D-21-19408R1

Dear Dr. Morrison,

We’re pleased to inform you that your manuscript has been judged scientifically suitable for publication and will be formally accepted for publication once it meets all outstanding technical requirements.

Kind regards,

Ambroise Wonkam

Academic Editor

PLOS ONE
---

## [Editor Report · Acceptance letter]

7 Oct 2021

PONE-D-21-19408R1 

External validation of the priapism impact profile in a Jamaican cohort of patients with sickle cell disease 

Dear Dr. Morrison:

I'm pleased to inform you that your manuscript has been deemed suitable for publication in PLOS ONE. Congratulations! Your manuscript is now with our production department. 

Kind regards, 

on behalf of

Professor Ambroise Wonkam 

Academic Editor

PLOS ONE